# The Effects of Environmental Factors on Ginsenoside Biosynthetic Enzyme Gene Expression and Saponin Abundance

**DOI:** 10.3390/molecules24010014

**Published:** 2018-12-20

**Authors:** Tao Zhang, Mei Han, Limin Yang, Zhongming Han, Lin Cheng, Zhuo Sun, Linlin Yang

**Affiliations:** Co-constructing Key Laboratory by Province and the Ministry of Science and Technology of Ecological Restoration and Ecosystem Management, College of Chinese Medicinal Material, Jilin Agricultural University, Changchun 130118, China; zt15584629802@126.com (T.Z.); hanzm2008@126.com (Z.H.); chenglin19880204@126.com (L.C.); 329575068@163.com (Z.S.); 13180883352@163.com (L.Y.)

**Keywords:** *Panax Ginseng*, gene expression, saponin, environmental factors, HPLC

## Abstract

*Panax ginseng* C.A. Meyer is one of the most important medicinal plants in Northeast China, and ginsenosides are the main active ingredients found in medicinal ginseng. The biosynthesis of ginsenosides is regulated by environmental factors and the expression of key enzyme genes. Therefore, in this experiment, ginseng in the leaf opened stage, the green fruit stage, the red fruit stage, and the root growth stage was used as the test material, and nine individual ginsenosides and total saponins (the sum of the individual saponins) were detected by HPLC (High Performance Liquid Chromatography). There was a trend of synergistic increase and decrease, and saponin accumulation and transfer in different tissues. The expression of key enzyme genes in nine synthetic pathways was detected by real-time PCR, and the correlation between saponin content, gene expression, and ecological factors was analyzed. Correlation analysis showed that in root tissue, PAR (Photosynthetically Active Radiation) and soil water potential had a greater impact on ginsenoside accumulation, while in leaf tissue, temperature and relative humidity had a greater impact on ginsenoside accumulation. The results provide a theoretical basis for elucidating the relationship between ecological factors and genetic factors and their impact on the quality of medicinal materials. The results also have guiding significance for realizing the quality of medicinal materials.

## 1. Introduction

*Panax ginseng* C.A. Meyer is a perennial herb that grows slowly [1], and has a long history of medicinal use in traditional Chinese medicine. Ginseng has numerous pharmacological effects on humans, including acting against tumors, as an antioxidant, to inhibit fat accumulation [2,3,4], improving immunity, improving erectile function, and to help combat cardiovascular and cerebrovascular disease [5,6,7,8,9,10,11]. These benefits come with few adverse reactions or side effects. According to the skeleton of aglycones, ginsenosides have been classified into two major types: dammarane and oleanane [12]. The dammarane-type ginsenosides contain two groups: protopanaxadiol (PPD) and protopanaxatriol (PPT) [13]. Ginseng grows very slowly and needs at least 5–6 years cultivation to be used in a medical clinic. Thus, it is widely accepted that the longer the ginseng grows, the better quality it is, with many previous studies reporting how the amount of total accumulated ginsenosides increases continuously with the duration of cultivation [14]. Analytical data on the accumulation and variation of ginsenoside levels in different tissues cultured at different growth stages, and the correlation between ecological factors and synthetic key enzyme gene expression are still missing.

Environmental factors affect the distribution, growth, yield, and quality of ginseng production areas. Ginseng medicinal materials have obvious local characteristics. Different ecological factors cause differences in the quality of the genuine regional drug. Ginseng grows best under some degree of shading. Shading of 20% is beneficial to the accumulation of ginsenoside content in ginseng roots, while 15% shading has been found to be beneficial for the accumulation of ginsenosides in the leaves of *Panax ginseng* [15,16]. A relative soil water content of 60–80% can promote the growth of ginseng; however, if the water content of the soil is too high, this can result in root decay due to lack of air in the soil pores. The relative humidity of the air affects plant growth via its effect on transpiration in the leaves [17,18]. At 400–600 m altitude, the ginsenoside content of plants increases with elevation above sea level. Ginseng volatile oil content varies greatly at different altitudes, but also generally increases with altitude [19]. 

In recent years, the ginsenoside biosynthesis pathway has been gradually clarified [20]. The biosynthesis pathway of ginsenosides is divided into two parts, upstream and downstream: (1) acetyl-CoA undergoes the action of mevalonate (MVA) to form 3-isoprene pyrophosphate (IPP) and dimethylallyl pyrophosphate (DMAPP), then IPP and DMAPP are catalyzed by various enzymes to form 2,3-oxidosqualene; (2) 2,3-oxidosqualene is cyclized, hydroxylated, and glycosylated, and structural modification is used to synthesize various monomeric ginsenosides, among which the key enzyme is the first in the MVA pathway in plants regulating ginsenoside synthesis is 3-hydroxy-3-methylglutaryl-CoA reductase (HMGR) [21]. A rate-limiting enzyme affects the production of isopentenyl pyrophosphate (IPP) and dimethylallyl pyrophosphate (DMAPP); farnesyl diphosphate synthase (FPS) [22] catalyzes the formation of farnesyl pyrophosphate (FPP) by IPP and geranyl pyrophosphate (GPP); squalene synthase (SS) [23] is involved in the catalytic synthesis of squalene; squalene epoxidase (SE) [24] catalyzes the formation of 2,3-oxidized squalene; dammarendiol synthase (DS) [25], and β-amyrin synthase (β-AS) [26] catalyze the formation of 2,3-oxidized squalene—a precursor of an alkane type, oleanane type ginsenoside—and finally, cytochrome P450 (CYP450) and glycosyl transferase (GT) undergo carbocyclic modification and glycosylation modification, and they exist in a plant in the form of a supergene family. This allows the formation of complex and diverse monomeric ginsenosides, and currently functionally verified genes such as CYP716A47 [27], CYP716A52v2 [28], and CYP716A53v2 [29]. However, most of the related research focuses on the exploration and functional verification of key enzyme genes, and there are few studies on the response mechanisms of plant internal genes and the external environment.

The unique natural geographical environment of Changbai Mountain in Jilin Province has formed the genuine regional drug area of ginseng medicine, and the synthesis of ginseng saponin is controlled by the growth environment and key enzyme genes in the body [30]. In this study, we used the HPLC method to measure nine individual ginsenosides. Through using this method, the characteristic components in the root, stem, and leaf of different growth stage ginseng were identified. Meanwhile, the content of the nine main ginsenosides in the three tissues of different growth times was compared. The distribution of these components in the inner tissues of ginseng was profiled, and the tissue expression characteristics of the nine key enzyme genes of the ginsenoside biosynthesis pathway were analyzed. Correlation analysis was carried out between ecological factors and gene expression levels and saponin content, to explore the response patterns of ginsenoside biosynthesis and gene expression to ecological factors. This provides a theoretical basis for elucidation of the physiological and ecological mechanisms of ginsenoside biosynthesis.

## 2. Results

### 2.1. Fresh Weight and Dry Weight of Roots

The important yield indexes of *Panax ginseng* are fresh and dry weights, and the study found that changes in the fresh and dry weights basically followed the same trend. In the different growth stages (leaf opened stage–root growth stage), the fresh and dry weights in the periods of green fruit were the lowest among the different growth stages. As shown in Figure 1, the fresh and dry weights of root gradually decreased from the leaf opened stage to the periods of green fruit, and then gradually increased from the green fruit stage to the periods of root growth stage. The fresh and dry weights increased significantly by the periods of root growth stage (*p* < 0.05) compared with the leaf opened stage. The fresh and dry weights of the roots had increased by 17.13% and 67.13% compared with the leaf opened stage.

### 2.2. Ginsenoside Content at Different Growth Times 

The ginsenoside content of the ginseng roots, stems and leaves was evaluated at different growth stages. As shown in Figure 2A and Figure 3A, among the roots and stems at different growth stages, the total ginsenoside content in the green fruit was the lowest at 24.488 mg/g and 7.296 mg/g, respectively. The highest total ginsenoside content appeared in the leaf opened stage, at 30.707 mg/g and 11.605 mg/g, and then increased slowly from the green fruit stage to the root growth stage. As shown in Figure 4A, at different growth stages of the leaf, the total ginsenoside content in the leaf opened stage was the lowest (141.726 mg/g) and the highest total ginsenoside content occurred in the root growth stage (211.825 mg/g), after increasing slowly from the leaf opened stage to the root growth stage. All the individual ginsenosides in the roots, stems, and leaves of different growth periods are shown in Figure 2B, Figure 3B and Figure 4B. The variation trend of most individual ginsenosides is the same as that of the total saponins.

According to the skeleton of aglycones, ginsenosides have been classified into two major types: dammarane and oleanane. Dammarane-type ginsenoside is the main type of ginsenoside. The dammarane-type ginsenosides contain two groups: protopanaxadiol (PPD) and protopanaxatriol (PPT). The changes of PPT-type (Rg1, Re, and Rf) and PPD-type (Rb1, Rb2, Rc, Rb3, and Rd) ginseng saponin content in different growth stages are shown in Figure 5.

The ratio of PPD-type ginsenosides to PPT-type ginsenosides changed in the different growth stages (Table 1). In the roots, in the transition from the leaf opened to the red fruit stage, this ratio increased, and the ratio decreased in the transition from the red fruit to the root growth stage. The ratio peaked during the red fruit period. In the transition from the leaf opened to the root growth stage, this ratio decreased in the leaf and stem. In particular, the PPD/PPT ratio in the stem was smaller than in the other organs at different growth times. Among the ginseng stems, PPT ginsenoside is relatively abundant, among which PPT ginsenoside content can reach five times the level of PPD ginsenoside content. The ratios of the main ginsenosides Rb1, Re, and Rg1 also changed in different organs during different growth times. In the roots, the percentage of the ginsenosides Rg1 and Re decreased; however, the percentage of Rb1 among the total ginsenosides increased. In the leaves, the ratio of Rg1 to the total ginsenosides increased during different growth times, and the percentage of the ginsenosides Re and Rb1 was maintained at a relatively stable level. In the stem, the percentage of ginsenoside Rg1, Re, and Rb1 did not change significantly. There was no significant change in Re and Rb1 in the leaves; it was maintained at a high content.

### 2.3. Biosynthesis of Ginsenoside-Related Genes at Different Growth Stages

To investigate the biosynthesis of ginsenosides, we conducted a real-time polymerase chain reaction to analyze the expression of the upstream and downstream synthetic pathway key enzyme gene in roots, stems, and leaves at different growth stages, including 3-hydroxy-3-methylglutaryl coenzyme A reductase (HMGR), farnesyl diphosphate synthase (FPS), squalene synthase (SS), squalene epoxidase (SE), dammarenediol synthase (DS), β-amyrin synthase (β-AS), and cytochrome P450 (*CYP716A47-PPDS*) (*CYP716A53v2-PPTS*) (*CYP716A52v2-OAS*). 

As shown in Figure 6, Figure 7 and Figure 8, expression of the upstream gene HMGR, FPS, SS and SE in the root and leaves rapidly decreased from the leaf opened to green fruit stage (78.5% and 58.9%, respectively, for HMGR; 61.2% and 80.4%, respectively, for FPS; 21.8% and 63.2%, respectively, for SS; 10.2% and 86.7%, respectively, for SE). The other genes showed similar expression patterns. The downstream genes DS, PPDS, and PPTS of the ginsenoside biosynthesis pathway were expressed more highly in leaves than other organs. On the contrary, β-AS and OAS were expressed more highly in roots than other organs. Interestingly, the expression pattern of the β-AS gene in roots at different growth stages is opposite to that of the OAS gene. It is speculated that there is a mutual inhibition between the two genes at different growth stages.

### 2.4. Environmental Factors 

The HOBO weather station was placed in the plot one month prior to the first sampling to collect meteorological data on the environmental conditions of the plot, including soil moisture, temperature, precipitation, and relative humidity. The data was averaged, as shown in the Table 2.

### 2.5. The Correlation between Individual Ginsenosides and Total Saponins

Correlation analysis results of individual ginsenosides and total saponins at different growth stages are shown in Table 3 and Table 4. In the roots, the abundance of Rf, Rb1, Rb2, and Rd was significantly correlated (*p* < 0.05) with the total saponins content. All individual ginsenosides, with the exception of the ginsenoside Ro, were positively correlated with total saponins. In the leaves, the abundance of PPD-type and PPT-type ginsenosides was significantly positively correlated (*p* < 0.05) with the total saponins content. However, only Ro was significantly negatively correlated with total saponins. Therefore, they also show that there is a synergistic trend between saponins, since they are significantly positively correlated. This is logical, because saponins share a common biosynthetic pathway. However, we also observed a significantly negative correlation between the monomer saponin Ro and total saponins, which may be due to the fact that Ro is an oleanane-type saponin. 

### 2.6. Correlation Analysis of Key Enzyme Gene Expression in the Ginsenoside Synthesis Pathway

The correlation analysis results of gene expression at different growth stages are shown in Table 5 and Table 6. In the roots, HMGR was significantly positively correlated (*p* < 0.01) with FPS, but it was significantly negatively correlated (*p* < 0.01) with DS and PPTS. SS was significantly positively correlated (*p* < 0.01) with SE. FPS was significantly negatively correlated (*p* < 0.01) with PPTS. DS was significantly negatively correlated (*p* < 0.01) with β-AS. β-AS was significantly negatively correlated (*p* < 0.01) with OAS. In the leaves, HMGR was significantly positively correlated (*p* < 0.01) with FPS. SS was significantly positively correlated (*p* < 0.01) with SE, PPDS, β-AS and OAS. SE was significantly positively correlated (*p* < 0.01) with PPDS, β-AS and DS. PPDS was significantly positively correlated (*p* < 0.01) with β-AS and OAS. β-AS was significantly positively correlated (*p* < 0.01) with OAS. Interestingly, PPTS was negatively correlated with all genes. The expression of nine key enzyme genes in ginseng is closely related with the others. The key enzyme genes are affected by the expression of several other key enzyme genes, which indicates that the biosynthesis process of ginsenosides in different tissues is affected. Synergistic regulation of multiple key enzymes promotes the expression of key enzyme genes, regulates the metabolic flow of ginsenosides, and ultimately guides the synthesis of various individual ginsenosides.

### 2.7. The Correlation between Environmental Factors and Ginsenosides and Gene Expression

The correlation analysis results of ecological factors and ginsenoside content are shown in Table 7 and Table 8. In the roots, PAR was significantly positively correlated (*p* < 0.01) with Rg1, Rf, and PPT-type ginsenosides. Soil water potential was negatively correlated with most of the individual ginsenosides. This indicated that PAR and soil water potential have certain effects on the biosynthesis of root ginsenosides. Soil water potential, relative humidity, and rain were negatively correlated with most ginsenosides, which inhibited the synthesis of those ginsenosides, indicating that the soil water potential, relative humidity, and rain were appropriately reduced, and suitable drought can improve the biosynthesis of ginsenosides. However, in the leaves, relative humidity was significantly positively correlated (*p* < 0.01) with all ginsenosides, except for the Ro and PPT-type ginsenosides. The results showed that in the leaves, relative humidity played an important role in the biosynthesis of ginsenosides, having a direct effect on ginseng leaves. The appropriate increase in relative humidity promoted the biosynthesis of ginsenosides.

The correlation analysis results of ecological factors and gene expression are shown in Table 9 and Table 10. In the roots, PAR was significantly positively correlated (*p* < 0.01) with HMGR and SS, but it was significantly negatively correlated with DS and PPTS. Relative humidity was significantly negatively correlated (*p* < 0.01) with HMGR, FPS, and β-AS, and it was positively correlated with PPTS and OAS. Soil water potential was significantly positively correlated with PPDS and β-AS. In the leaves, relative humidity and rain play a large role. Relative humidity and rain were negatively correlated with most of the genes. 

In the roots, HMGR was significantly positively correlated (*p* < 0.01) with Re, Rf, and Rb3 (Table 11). SS was significantly positively correlated (*p* < 0.01) with Rg1 and PPT-type ginsenosides. FPS was significantly positively correlated (*p* < 0.01) with Re. SE was significantly positively correlated (*p* < 0.01) with PPT-type ginsenosides. DS was significantly positively correlated (*p* < 0.01) with Re, Rf and PPT-type ginsenosides. SS, FPS, SE, DS, PPDS, β-AS, and OAS gene expression was significantly correlated with ginsenosides. It is indicated that SS, FPS, SE, DS, PPDS, β-AS, and OAS genes have synergistic regulation roles in leaves (Table 12). The expression of a key enzyme gene in ginsenoside synthesis may only act on a certain site in the ginsenoside synthesis pathway. Due to multiple regulatory sites in the ginsenoside synthesis pathway. There are multiple steps between saponin synthesis, so not all genes have a significant correlation with saponin content.

## 3. Discussion

The quality of medicinal materials when they form is affected by the external environment. Changes in the external environment lead to changes in the genetic factors of medicinal materials. The biosynthesis of medicinal ingredients reflects the diversity of responses to environmental factors. The internal genetic factors and external environmental factors determine the quality of medicinal materials. Ginseng saponin, the main active ingredient of ginseng, has many pharmacological and immunological activities [31,32,33]. Ginseng modestly yet significantly improves fasting blood glucose in people with and without diabetes [34], and a study by Lin et al. showed that ginseng also enhances cardiac contractility in other animal species [35].

Environmental factors affect the distribution, growth, yield, and quality of ginseng production areas. Ginseng medicinal materials have obvious local characteristics. Different ecological factors cause differences in the quality of genuine regional drugs. The main ecological factors affecting the quality of ginseng medicinal materials are temperature, relative humidity, rain, moisture, altitude, soil physical qualities, and PAR. A higher ginsenoside content of leaves may be the result of a positive correlation with light, which would agree with prior observations that higher levels of light transmission increase ginsenosides in ginseng plant leaves [36]. 

Ginsenoside content is related to the expression of key ginsenoside biosynthesis genes during foliation, as was shown in a study of 3-year-old *Panax ginseng* Meyer [37]. These results suggest that PPD-type and PPT-type ginsenosides are produced according to growth stage to meet different needs in the growth and defense of ginseng. The results provide support for the accumulation and changes of PPT-type and PPD-type ginsenosides at different growth stages. PPT ginsenoside has a high content in ginseng leaves and PPD ginsenoside has a high content in ginseng roots, which provides a basis for the molecular explanation that PPD ginsenoside contained in roots is partly derived from the synthesis and transportation processes of leaves.

In this study, PAR was significantly positively correlated with Rg1, Rf, and PPT-type ginsenosides. The ginsenoside content of ginseng roots increased as the light transmission rate (LTR) increased. A study of 2-year-old ginseng reported that the total ginsenoside content of ginseng grown at 17% LTR was 49.7% and 68.3% higher than ginseng grown at 6% LTR in August and at final harvest, respectively [38]. These results are consistent with the results of this study. This indicates that an appropriate PAR was beneficial to the accumulation of ginsenosides. In the leaves, relative humidity was significantly positively correlated with all ginsenosides, except for the Ro and PPT-type ginsenosides. Relative humidity can appropriately alleviate the drought limit of the soil, which is conducive to plant leaf growth and life activities. Ginseng leaves are sensitive to light, and excessive photosynthetically active radiation may damage the leaves, thus causing the reduction of ginsenosides. Soil water potential were negatively correlated with most of the individual ginsenosides, and suitable drought conditions can improve the biosynthesis of ginsenosides. Relative humidity is an important factor affecting the quality of medicinal materials. Appropriate drought stress can improve the active ingredients in these medicinal materials [39].

With the development of molecular biology, people have also begun to focus on the influence of intrinsic genetic factors on the formation of medicinal materials [40]. HMGR was differentially expressed among tissues, with a high level of expression in the leaf and low level of expression in the stem, suggesting that leaves are crucial to terpenoid biosynthesis [41]. Kim et al. reported that overexpression of FPS caused an approximate 2.4-fold increase of ginsenoside content in transgenic ginseng hairy roots [42]. Seo et al. introduced the SS gene of P. ginseng into Eleutherococcus senticosus by Agrobacterium-mediated transformation, and found that SS activity significantly improved the production of phytosterols and triterpenoids [43]. He et al. cloned SQE from the root of P. notoginseng by PCR. Real time quantitative PCR analysis showed that its cDNA had a different expression pattern and is highly expressed in the root, especially in three-year-old root [44]. Chen et al. analyzed the transcriptomes of *P. ginseng* and identified 133 CYP450 genes by 454 sequencing technology. Their study laid an important foundation for the further screening of CYP450 involved in ginsenoside biosynthesis [45].

In this study, expression analysis by real-time quantitative PCR indicated that genes were differentially expressed among tissues. SS, DS, SE, PPTS, and PPDS had high levels of expression in the leaf, which is closely related to the distribution of dammarane ginsenoside in leaves. FPS, HMGR, β-AS, and OAS had high levels of expression in the root, which is consistent with the distribution of Ro mainly in the root. Through the analysis of gene correlation, it was found that there was a significant correlation between the expression of multiple enzyme genes in the pathway, indicating that multiple ginsenoside synthesis genes in the pathway interacted or had the same regulatory element, which was a key gene in ginsenoside biosynthesis. Through correlation analysis of gene expression and saponins content, it was found that multiple enzymes in the pathway were significantly correlated with single saponins at different growth stages, and thus played different regulatory roles.

To the best of our knowledge, information about the correlation between changes in ginseng saponin content and the expression of key biosynthesis genes and environmental factors at different growth times has not been reported. In this study we used 5-year-old ginseng plants for ginsenoside analysis. The variation trend of most individual ginsenosides is the same as that of the total saponins. Among the roots and stems at different growth stages, the total ginsenoside content decreased rapidly from the leaf opened stage to the green fruit stage, and then slowly increased from the green fruit stage to the root growth stage. At different growth stages of the leaves, the ginsenosides from the leaf opened stage to the root growth phase slowly increased. This may be due to the need for ginseng reproductive growth during the opening period of the leaves, which consumes nutrients stored in the roots, as shown by Lee, S.W. et al. [46]. We speculate that since roots serve as organs for plant nutrient storage, as plants consume nutrients stored in the roots during the different growth times, the metabolites already stored in the roots from the last season might be transported to parts of the plant above ground. Therefore, this may be the cause of the rapid decrease of ginsenoside content in the green fruit stage. In later stages, however, nutrients continue to accumulate in the roots [20,37]. The content of PPD-type and PPT-type ginsenosides in leaves was higher than that in other organs. PPT-type ginsenosides in stems were significantly higher than PPD-type ginsenosides, and PPD-type ginsenosides in roots were higher than PPT-type ginsenosides. Previous studies have shown that PPD-type ginsenosides are mainly distributed in roots, and PPT-type ginsenosides are mainly distributed in leaves. The results of gene expression have proven the conclusions of previous studies [20,21,24], suggesting that the precursors of transferred ginsenosides are mostly PPD-type ginsenosides.

## 4. Materials and Methods

### 4.1. Plant Material

Samples of fresh roots, stem, leaves, and berries from 5-year-old Ginseng plants were obtained from plants growing in the LaoLing plot (N41°54′37″, E127°41′27″), located in Fusong County, Jilin Province, China. It was identified as *Panax ginseng* C.A. Meyer by Professor Yang Limin from the College of Chinese Herbal Medicine of Jilin Agricultural University. Roots, stem, leaves, and berries were sampled at different growth stages (leaf opened, green fruit, red fruit, and root growth stage) (Figure 9). During leaf unfolding, curled leaves unfold gradually to become entirely flat. Smooth leaves lack wrinkles, and vary in color from dark green to matte yellow-green. At first leaf growth is very slow, and this picks up over time. During the green fruit stage, withered petals undergo abscission, ovaries gradually expand, and small fruit appear along the inflorescence. During this period, the plant reaches peak consumption of water and soil nutrients, as well as peak photosynthetic activity. This increased activity is presumably required to meet the energetic and nutrient needs of growing fruit and storage roots. During the red fruit stage, fruit enlarge, and change in color twice, first from green to purple, then from purple to red. During the root growth stage, in order to provide nutrients for the growth of roots, stems, and leaves, organic matter produced in the stems and leaves is transported to the underground organs for storage. This growth time is specific to perennial herbs, as annuals die before overwintering. A portion of plant materials were immediately frozen in liquid nitrogen upon harvesting and stored at –80 °C for RNA isolation, and the other portion was dried at 60 °C for seven days and used for ginsenoside analysis. Three independent biological replicates were prepared and each replicate included material from three or more plants.

### 4.2. Determination of Fresh and Dry Weight of Panax Ginseng

The roots of *Panax ginseng* were washed with tap water and rinsed three times with distilled water, then the surface moisture was wiped off the roots with filter paper. The fresh weight (FW) was recorded, the roots stem, leaves, and berries were oven dried at 60 °C for 7 days, and then the dry weight (DW) was recorded.

### 4.3. Analysis of Ginsenosides by HPLC

Milled powder from heat-dried roots was separated using a 60-mesh sieve, and 1.0 g milled powder was weighed once from each of three technical replicates. Ginsenosides were extracted using a microwave protocol. Extraction used the following conditions: 600 W extraction power, 45 °C extraction temperature, and 5 min extraction time, with a material: liquid ratio of 1:20. Methanol extraction was performed three times, then samples were filtered in a 25 mL conical flask, followed by rotary evaporation to a 10 mL capacity bottle. It was then passed through a 0.22 μm filter for HPLC analysis. HPLC separation was carried out on an Agilent 1260 series HPLC system (Agilent, Palo Alto, CA, USA), equipped with an autosampler and an UV detector using a C18 column (4.6 mm × 250 mm, 5 μm; Agilent). Gradient elution was preformed using solvent A (100% acetonitrile) and solvent B (100% water) at 25 °C, using the following gradient program (all isocratic): 0–18 min, 18–23% A; 18–28 min, 23–28% A; 28–31.1 min, 28–31% A; 31.1–49 min, 31–36% A; 49–68 min, and 36–82% A. The flow rate was maintained at 1.0 ml/min, the sample injection volume was 10 μL, and UV absorption was measured at 203 nm. Quantitative analysis was performed using the one-point curve method using external standards of authentic ginsenosides. Total saponins were determined by UV spectrophotometry.

A standard curve was established using the ginsenosides standard sample Re, Rg1, Rf, Rb1, Rb2, Rb3, Rc, Rd, and Ro. High-performance liquid chromatography (HPLC) was used to determine ginsenoside content, which was calculated according to the standard curve equation (Table 13). 

### 4.4. Extraction of RNA and Quantification of Transcript Levels

Total RNA was extracted from the frozen samples using the TaKaRa MiniBEST Universal RNA Extraction Kit. A P330 nanophotometer was used to determine RNA purity and concentration. The PrimeScript^TM^ RT Master Mix kit was used for first strand cDNA synthesis. Real-time quantitative polymerase chain reaction amplification was performed using a reaction volume of 20.0 μL, which contained 1.0 μL cDNA, 10.0 μL SYBR Green Premix Ex Taq^TM^, 1.0 μL each forward and reverse primer, and 7.0 μL d_2_H_2_O. The thermal cycler conditions recommended by the manufacturer were used: 30 s at 95 °C, followed by 40 cycles at 95 °C for 5 s, 55 °C for 32 s, and 72 °C for 20 s. The fluorescent product was detected during the final step of each cycle. Amplification, detection, and data analysis were carried out using an Agilent Technologies Stratagene Mx3000P thermocycler (Agilent, Palo Alto, CA, USA). To determine the relative fold-differences in template abundance for each sample, the Ct values for each of the gene-specific primers were normalized to the Ct value for *GAPDH* and calculated relative to a calibrator using the formula 2^−ΔΔCt^. Primer sequence design is as shown in Table 14.

### 4.5. Environmental Factor Data Collection

In this experiment, a HOBO meteorological station was used to collect the meteorological data of the environmental conditions in the LaoLing plot, including soil water content, temperature, precipitation, and relative humidity. The instrument is used for 24 h dynamic monitoring, and data points were collected once every half hour.

### 4.6. Statistical Analysis

The relative expression of the target genes was analyzed using the 2^−ΔΔCt^ method [47]. Excel 2016 was used to arrange the raw data. SPSS 19.0 (IBM Corporation, Armonk, NY, USA) was used for single-factor variance analysis and Pearson correlation statistical analysis. GraphPad Prism 7.0 was used to graph the data. 

## 5. Conclusions

The ginsenoside content of ginseng grown on site was analyzed by HPLC for the first time, and the expression levels of key enzyme genes in nine synthetic pathways were detected. The accumulation of ginsenosides in different tissues and organs is metastatic. A synergistic promotion or inhibition between saponins was found. Synthetic pathway key enzyme gene expression levels also have synergistic promotion or inhibition. Correlation analysis showed that in root tissue, PAR and soil water potential had a greater impact on ginsenoside accumulation, while in leaf tissue, temperature and relative humidity had a greater impact on ginsenoside accumulation. The results provide a theoretical basis for elucidating the relationship between ecological factors and genetic factors and the quality of medicinal materials, and have guiding significance for achieving the regulation of the quality of medicinal materials.

## Figures and Tables

**Figure 1 molecules-24-00014-f001:**
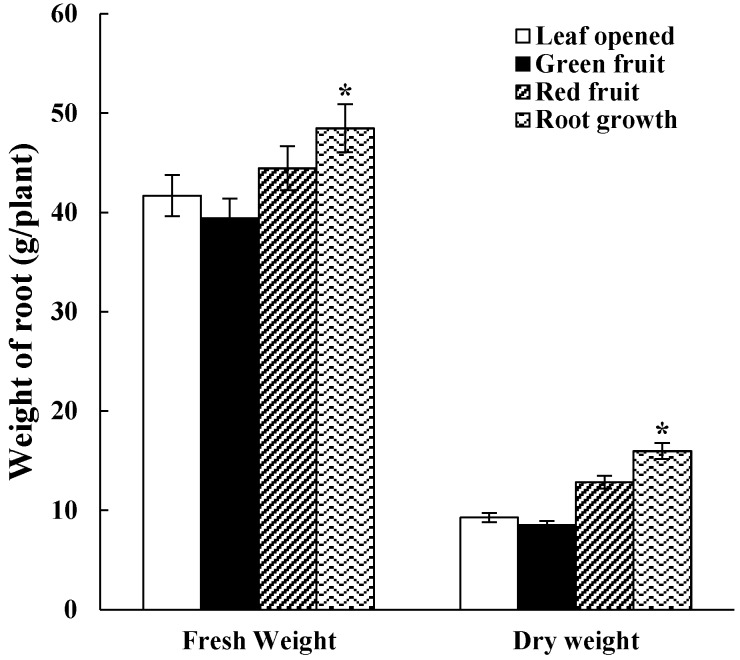
Fresh and dry weights of ginseng roots at different growth stages. The data are expressed as the mean ± SD. * indicated that the different growth stages are significantly different at the 0.05 level.

**Figure 2 molecules-24-00014-f002:**
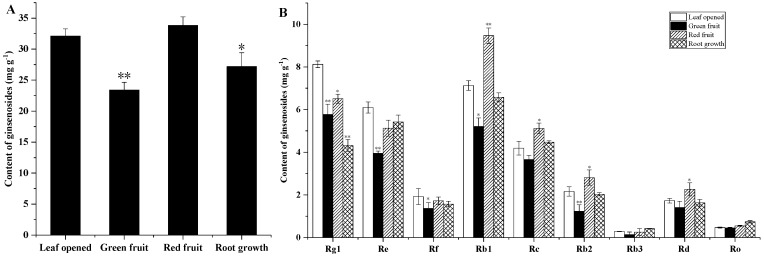
Ginsenoside content of roots from *Panax ginseng* during the different growth stages. The total ginsenoside content in the root (**A**) and the major individual ginsenosides in the root (**B**) were analyzed according to the leaf opened, green fruit, red fruit, and root growth stages. Vertical bars indicate the mean ± standard error from three independent experiments. The data are expressed as the mean ± SD. * indicated that the different growth stages are significantly different at the 0.05 level. ** indicated that the different growth stages are significantly different at the 0.01 level.

**Figure 3 molecules-24-00014-f003:**
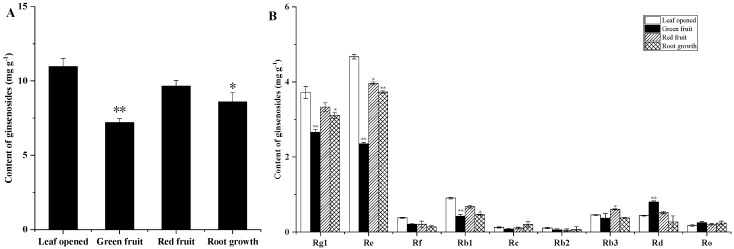
Ginsenoside contents of stems from *Panax ginseng* during the different growth stages. The total ginsenoside content in the stem (**A**) and the major individual ginsenosides in the stem (**B**) were analyzed according to the leaf opened, green fruit, red fruit, and root growth stages. Vertical bars indicate the mean ± standard error from three independent experiments. The data are expressed as the mean ± SD. * indicated that the different growth stages are significantly different at the 0.05 level. ** indicated that the different growth stages are significantly different at the 0.01 level.

**Figure 4 molecules-24-00014-f004:**
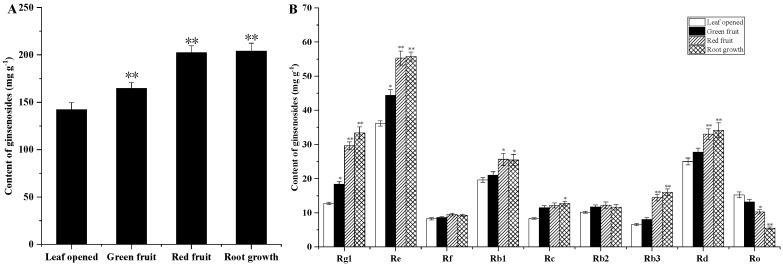
Ginsenoside contents of leaves from *Panax ginseng* during the different growth stages. The total ginsenoside content in the leaf (**A**) and the major individual ginsenosides in the leaf (**B**) were analyzed according to the leaf opened, green fruit, red fruit, and root growth stages. Vertical bars indicate the mean ± standard error from three independent experiments. The data are expressed as the mean ± SD. * indicated that the different growth stages are significantly different at the 0.05 level. ** indicated that the different growth stages are significantly different at the 0.01 level.

**Figure 5 molecules-24-00014-f005:**
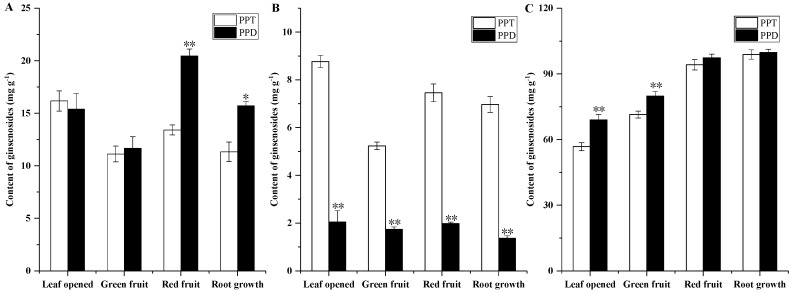
The ginsenoside compositions of the root (**A**), leaf (**B**), and stems (**C**) in the different growth stages. Vertical bars indicate the mean ± standard error from three independent experiments. * indicated that the different growth stages are significantly different at the 0.05 level. ** indicated that the different growth stages are significantly different at the 0.01 level.

**Figure 6 molecules-24-00014-f006:**
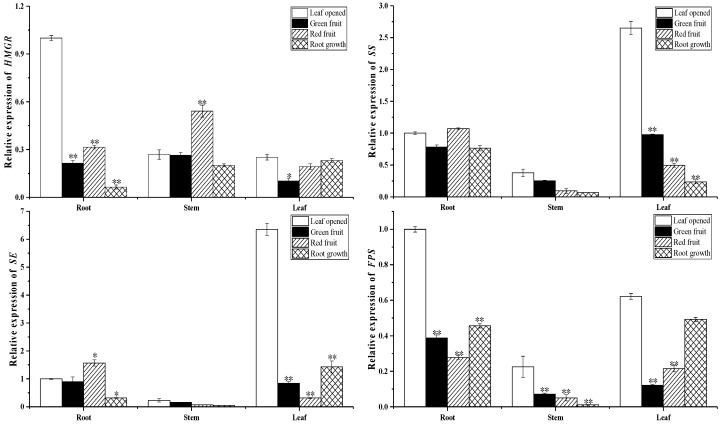
Expression of genes related to ginsenoside biosynthesis in roots, stems, and leaves during different growth times. The relative expression of (**A**) HMGR, (**B**) SS, (**C**) SE, and (**D**) FPS genes in the leaf opened, green fruit, red fruit, and root growth stages was analyzed by real-time polymerase chain reaction. Vertical bars indicate the mean ± standard error from three independent experiments. * indicated that the different growth stages are significantly different at the 0.05 level. ** indicated that the different growth stages are significantly different at the 0.01 level.

**Figure 7 molecules-24-00014-f007:**
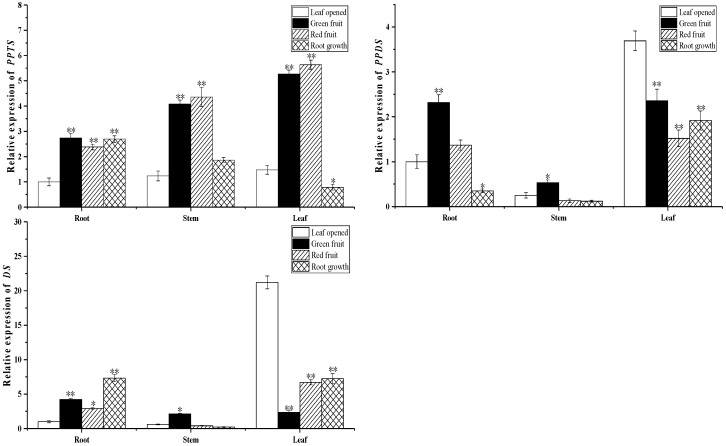
Expression of genes related to ginsenoside biosynthesis in roots, stems, and leaves during different growth times. The relative expression of (**A**) PPTS, (**B**) PPDS, and (**C**) DS genes in the leaf opened, green fruit, red fruit, and root growth stages was analyzed by real-time polymerase chain reaction. Vertical bars indicate the mean ± standard error from three independent experiments. * indicated that the different growth stages are significantly different at the 0.05 level. ** indicated that the different growth stages are significantly different at the 0.01 level.

**Figure 8 molecules-24-00014-f008:**
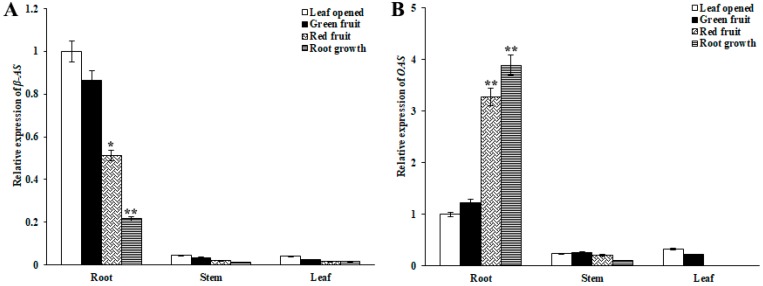
Expression of genes related to ginsenoside biosynthesis in roots, stems, and leaves during different growth times. The relative expression of (**A**) β-AS and (**B**) OAS genes in the leaf opened, green fruit, red fruit, and root growth stages was analyzed by real-time polymerase chain reaction. Vertical bars indicate the mean ± standard error from three independent experiments. * indicated that the different growth stages are significantly different at the 0.05 level. ** indicated that the different growth stages are significantly different at the 0.01 level.

**Figure 9 molecules-24-00014-f009:**
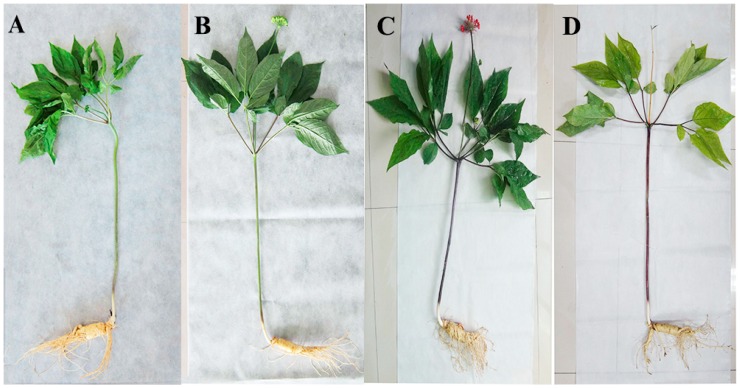
Growth stage of ginseng. Five-year-old ginseng plants were sampled. For the ginsenoside analysis and RNA extraction, the root, stem, and leaf were sampled at different growth stages, including the (**A**) leaf opened, (**B**) green fruit, (**C**) red fruit and (**D**) root growth stages.

**Table 1 molecules-24-00014-t001:** The ginsenoside compositions of the root, stem, and leaf during different growth times.

Ginsenosides	Root				Stem				Leaf			
	LO	GF	RF	RG	LO	GF	RF	RG	LO	GF	RF	RG
PPD/PPT	0.95	1.05	1.52	1.39	0.23	0.33	0.27	0.20	1.22	1.12	1.03	1.01
Rg1/total ginsenosides	0.25	0.25	0.19	0.16	0.34	0.37	0.35	0.36	0.09	0.11	0.14	0.16
Re/total ginsenosides	0.19	0.17	0.15	0.20	0.43	0.33	0.41	0.43	0.25	0.26	0.27	0.26
Rb1/total ginsenosides	0.22	0.24	0.28	0.24	0.08	0.06	0.07	0.05	0.14	0.12	0.12	0.12

**Table 2 molecules-24-00014-t002:** Ecological factor data in different growth times.

	Temperature (°C)	PAR (μE)	Relative Humidity (%)	Rain (mm)	Soil Water Potential (J/g)
Leaf opened	14.961c	531.67a	65.289c	19.333d	−0.178b
Green fruit	16.845b	365.89c	81.067b	44.519b	−0.108c
Red fruit	23.174a	499.28b	85.134a	51.267a	−0.223a
Root growth	15.314c	343.17d	86.609a	24.516c	−0.289a

Lowercase letters after data indicate significant differences in meteorological data during each growth times (*p* < 0.05).

**Table 3 molecules-24-00014-t003:** Correlation analysis of individual ginsenosides and total saponins in roots at different growth times.

	PPT			PPD					Ro	Total Saponins
	Rg1	Re	Rf	Rb1	Rc	Rb2	Rb3	Rd		
Rg1	1	0.43	0.73 *	0.31	−0.10	0.27	−0.36	0.29	−0.75 *	0.58
Re		1	0.91 *	0.44	0.42	0.61	0.69 *	0.36	0.27	0.72 *
Rf			1	0.61	0.46	0.69 *	0.35	0.54	−0.09	0.88 **
Rb1				1	0.95 **	0.97 **	0.22	0.99 **	0.13	0.91 **
Rc					1	0.96 **	0.45	0.94 **	0.43	0.80 *
Rb2						1	0.43	0.95 **	0.27	0.94 **
Rb3							1	0.15	0.88 *	0.28
Rd								1	0.09	0.88 *
Ro									1	−0.01
Total saponins										1

* indicates a significant correlation (*p* < 0.05). ** indicates a significant correlation (*p* < 0.01).

**Table 4 molecules-24-00014-t004:** Correlation analysis of individual ginsenosides and total saponins in leaves at different growth times.

	PPT			PPD					Ro	Total Saponins
	Rg1	Re	Rf	Rb1	Rc	Rb2	Rb3	Rd		
Rg1	1	0.98 **	0.95 **	0.98 **	0.89 **	0.73 *	0.99 **	1.00 **	−0.95 **	0.99 **
Re		1	0.98 **	0.98 **	0.94 **	0.84 *	0.97 **	0.99 **	−0.89 **	1.00 **
Rf			1	0.99 **	0.89 **	0.85 **	0.94 **	0.97 **	−0.81 *	0.99 **
Rb1				1	0.86 **	0.76 *	0.99 **	0.99 **	−0.87 **	0.99 **
Rc					1	0.91 **	0.84 *	0.90 **	−0.83 *	0.92 **
Rb2						1	0.67	0.76 *	−0.56	0.81 *
Rb3							1	0.99 **	−0.94 **	0.98 **
Rd								1	−0.93 **	0.99 **
Ro									1	−0.89 **
Total saponins										1

* indicates a significant correlation (*p* < 0.05). ** indicates a significant correlation (*p* < 0.01).

**Table 5 molecules-24-00014-t005:** Correlation analysis of gene expression in roots at different growth times.

	HMGR	SS	FPS	SE	DS	PPDS	PPTS	β-AS	OAS
HMGR	1	0.59	0.89 **	0.31	−0.87 **	−0.06	−0.99 **	0.74 *	−0.67 *
SS		1	0.21	0.85 **	−0.78 *	−0.02	−0.58	0.27	−0.08
FPS			1	−0.16	−0.55	−0.29	−0.92 **	0.59	−0.59
SE				1	−0.72 *	0.46	−0.23	0.36	−0.19
DS					1	−0.32	0.79 *	−0.81 **	0.68 *
PPDS						1	0.22	0.59	−0.60
PPTS							1	−0.63	0.55
β-AS								1	−0.98 **
OAS									1

* indicates a significant correlation (*p* < 0.05). ** indicates a significant correlation (*p* < 0.01).

**Table 6 molecules-24-00014-t006:** Correlation analysis of gene expression in leaves at different growth times.

	HMGR	SS	FPS	SE	DS	PPDS	PPTS	β-AS	OAS
HMGR	1	0.32	0.91 **	0.61	0.77 *	0.33	−0.77 *	0.27	−0.06
SS		1	0.54	0.93 **	0.85 *	0.96 **	−0.28	0.99 **	0.90 **
FPS			1	0.81 *	0.85 *	0.62	−0.92 **	0.54	0.26
SE				1	0.96 **	0.95 **	−0.60	0.93 **	0.75 *
DS					1	0.82 *	−0.59	0.81 *	0.56
PPDS						1	−0.46	0.99 **	0.92 **
PPTS							1	−0.33	-0.12
β-AS								1	0.94 **
OAS									1

* indicates a significant correlation (*p* < 0.05). ** indicates a significant correlation (*p* < 0.01).

**Table 7 molecules-24-00014-t007:** Correlation analysis between environmental factors and ginsenosides in roots at different growth times.

	Temperature (°C)	PAR (μE)	Relative Humidity (%)	Rain (mm)	Soil Water Potential (J/g)
Rg1	0.07	0.91 **	−0.85 **	−0.13	0.45
Re	−0.23	0.58	−0.51	−0.71 *	−0.55
Rf	0.03	0.87 **	−0.65	−0.46	−0.26
Rb1	0.77 *	0.66	0.12	0.31	−0.45
Rc	0.71 *	0.41	0.36	0.26	−0.69 *
Rb2	0.62	0.64	0.07	0.10	−0.60
Rb3	−0.28	−0.12	0.16	−0.62	−0.94 **
Rd	0.83 **	0.64	0.17	0.40	−0.40
Ro	−0.10	−0.49	0.60	−0.28	−0.92 **
PPT	−0.03	0.93 **	−0.84 **	−0.40	0.07
PPD	0.73 *	0.54	0.23	0.26	−0.59
Total saponins	0.60	0.76 *	−0.08	0.08	−0.48

* indicates a significant correlation (*p* < 0.05). ** indicates a significant correlation (*p* < 0.01).

**Table 8 molecules-24-00014-t008:** Correlation analysis between environmental factors and ginsenosides in leaves at different growth times.

	Temperature (°C)	PAR (μE)	Relative Humidity (%)	Rain (mm)	Soil Water Potential (J/g)
Rg1	0.40	−0.44	0.89 **	0.26	−0.79 *
Re	0.53	−0.43	0.94 **	0.43	−0.67
Rf	0.66	−0.27	0.89 **	0.51	−0.64
Rb1	0.56	−0.28	0.86 **	0.36	−0.76 *
Rc	0.41	−0.67	1.00 **	0.52	−0.44
Rb2	0.70	−0.43	0.93 **	0.83 **	−0.17
Rb3	0.41	−0.35	0.83 **	0.21	−0.84 **
Rd	0.45	−0.24	0.89 **	0.31	−0.77
Ro	−0.09	0.60	−0.81 **	0.00	0.83 **
PPT	−0.03	0.93 **	−0.84 **	−0.40	0.07
PPD	0.73 *	0.54	0.23	0.26	−0.59
Total saponins	0.73 *	0.04	0.69 *	0.42	−0.72 *

* indicates a significant correlation (*p* < 0.05). ** indicates a significant correlation (*p* < 0.01).

**Table 9 molecules-24-00014-t009:** Correlation analysis between environmental factors and gene expression in roots at different growth times.

	Temperature (°C)	PAR (μE)	Relative Humidity (%)	Rain (mm)	Soil Water Potential (J/g)
HMGR	−0.24	0.82 **	−0.96 **	−0.47	0.30
SS	0.62	0.94 **	−0.35	0.21	−0.02
FPS	−0.64	0.50	−0.94 **	−0.81 **	0.13
SE	0.82 **	0.73 *	−0.12	0.65	0.33
DS	−0.19	−0.91 **	0.78 *	−0.03	−0.52
PPDS	0.27	−0.03	−0.02	0.67	0.91 **
PPTS	0.28	−0.81 **	0.93 **	0.57	−0.13
β-AS	−0.21	0.50	−0.82 **	−0.07	0.86 **
OAS	0.36	−0.33	0.79 *	0.14	−0.88 **

* indicates a significant correlation (*p* < 0.05). ** indicates a significant correlation (*p* < 0.01).

**Table 10 molecules-24-00014-t010:** Correlation analysis between environmental factors and gene expression in leaves at different growth times.

	Temperature (°C)	PAR (μE)	Relative Humidity (%)	Rain (mm)	Soil Water Potential (J/g)
HMGR	−0.24	0.44	−0.37	−0.72 *	−0.69 *
SS	−0.43	0.65	−1.00 **	−0.52	0.46
FPS	−0.60	0.31	−0.58	−0.94 **	−0.46
SE	−0.59	0.57	−0.95 **	−0.78 *	0.12
DS	−0.38	0.71 *	−0.85 **	−0.72 *	−0.07
PPDS	−0.66 *	0.43	−0.97 **	−0.69 *	0.41
PPTS	0.77 *	0.10	0.33	0.96 **	0.54
β-AS	−0.55	0.53	−0.99 **	−0.58	0.49
OAS	−0.54	0.34	−0.89 **	−0.38	0.74 *

***** indicates a significant correlation (*p* < 0.05). ** indicates a significant correlation (*p* < 0.01).

**Table 11 molecules-24-00014-t011:** Correlation analysis between gene expression and ginsenosides in roots at different growth times.

	HMGR	SS	FPS	SE	DS	PPDS	PPTS	β-AS	OAS
Rg1	0.25	0.87 **	0.28	0.71 *	0.74 *	0.70 *	0.10	0.79 *	0.69 *
Re	0.98 **	0.48	0.91 **	0.72 *	0.87 **	0.46	−0.71 *	0.42	0.09
Rf	0.82 **	0.63	0.72 *	0.74 *	0.90 **	0.51	−0.39	0.54	0.24
Rb1	0.43	−0.12	0.04	−0.09	0.19	−0.34	0.23	−0.27	−0.47
Rc	0.47	−0.37	0.06	−0.25	0.04	−0.53	0.09	−0.49	−0.71 *
Rb2	0.61	−0.08	0.25	0.03	0.32	−0.26	0.00	−0.22	−0.48
Rb3	0.81 **	−0.21	0.71 *	0.17	0.30	−0.10	−0.80 **	−0.21	−0.47
Rd	0.34	−0.16	−0.05	−0.16	0.12	−0.39	0.31	−0.31	−0.49
Ro	0.44	−0.64	0.30	−0.31	−0.20	−0.51	−0.53	0.62	0.77 *
PPT	0.62	0.83 **	0.60	0.83 **	0.92 **	0.69 *	−0.24	0.74 *	0.52
PPD	0.48	−0.24	0.08	−0.16	0.14	−0.43	0.13	−0.38	−0.60
Total saponins	0.61	0.07	0.27	0.14	0.42	−0.13	0.03	−0.07	−0.34

* indicates a significant correlation (*p* < 0.05). ** indicates a significant correlation (*p* < 0.01).

**Table 12 molecules-24-00014-t012:** Correlation analysis between gene expression and ginsenosides in leaves at different growth times.

	HMGR	SS	FPS	SE	DS	PPDS	PPTS	B-AS	OAS
Rg1	0.11	−0.90 **	−0.18	−0.71 *	−0.55	−0.88 **	−0.01	−0.92 **	−0.99 **
Re	−0.04	−0.95 **	−0.34	−0.81 **	−0.65	−0.95 **	0.17	−0.97 **	−0.99 **
Rf	−0.03	−0.90 **	−0.37	−0.80 **	−0.60	−0.95 **	0.26	−0.95 **	−0.99 **
Rb1	0.12	−0.87 **	−0.22	−0.71 *	−0.51	−0.90 **	0.10	−0.92 **	−1.00 **
Rc	−0.34	−1.00 **	−0.54	−0.93 **	−0.86 **	−0.95 **	0.28	−0.98 **	−0.89 **
Rb2	−0.55	−0.91 **	−0.80 **	−0.99 **	−0.90 **	−0.97 **	0.65	−0.93 **	−0.79 **
Rb3	0.21	−0.85 **	−0.09	−0.63	−0.45	−0.83 **	−0.06	−0.88 **	−0.98 **
Rd	0.08	−0.91 **	−0.21	−0.73 *	−0.56	−0.90 **	0.04	−0.93 **	−0.99 **
Ro	−0.19	0.84 **	−0.01	0.58	0.47	0.73 *	0.27	0.82 **	0.89 **
PPT	0.62	0.83 **	0.60	0.83 **	0.92 **	0.69	−0.24	0.74 *	0.52
PPD	0.48	−0.24	0.08	−0.16	0.14	−0.43	0.13	−0.38	−0.60
Total saponins	0.22	−0.70 *	−0.18	−0.58	−0.32	−0.81 **	0.19	−0.79 *	−0.92 **

* indicates a significant correlation (*p* < 0.05). ** indicates a significant correlation (*p* < 0.01).

**Table 13 molecules-24-00014-t013:** Regression equations with correlation coefficients of ginsenoside monomers.

−	Standard Curve Equation	R^2^	Linear Range (μg)
Ro	y = 2796.01284 x − 40.64132	0.99969	6.0~0.375
Rg_1_	y = 3131.42174 x − 34.75556	0.99943	6.0~0.375
Re	y = 3097.80287 x − 35.99583	0.99945	6.0~0.375
Rf	y = 3614.44564 x − 32.36528	0.99912	6.0~0.375
Rb_1_	y = 2246.85066 x − 17.05278	0.99989	6.0~0.375
Rc	y = 2528.59020 x − 19.43056	0.99985	6.0~0.375
Rb_2_	y = 2668.61410 x − 36.21944	0.99941	6.0~0.375
Rb_3_	y = 3451.35006 x − 47.50556	0.99932	6.0~0.375
Rd	y = 3009.68339 x − 30.33472	0.99967	6.0~0.375

**Table 14 molecules-24-00014-t014:** Sequences of primers for real-time RT-PCR.

Gene Name	Accession No.	Primer (5′−3′)	Reference
*GAPDH*	KF699323	F: ATGGACCATCAGCAAAGGAC R: GGTAGCACTTTCCCAACAGC	Liu J et al. [48]
*HMGR*	GQ455990	F: TTGGATTGAAGGGCGAGGAAAG R: CAGCAACAGCAGAACCAGCAAG	Kim et al. (2014b) [21]
FPS	DQ087959	F: CAAGAAGCATTTCCGACAA R: CTCTCCTACAAGGGTGGTGA	Kim et al. (2010) [22]
SS	AB115496	F: GGACTTGTTGGATTAGGGTTG R: ACTGCCTTGGCTGAGTTTTC	Lee et al. (2004) [23]
SE	AB122078	F: ATGCTTTGAATATGCGCCATC R: CATGGAGATCGCGTAAAGGTC	Han et al. (2010) [24]
DS	AB122080	F: ACCGCCGTTGAGATTAGATG R: ATAGGGCAATGATAAGGGGAG	Han et al. (2006) [25]
β-AS	AB009030	F: GCGGAAGGGAATAAGATGAC R: CTCAGCTCTCCGGACAGC	Kushiro et al. (1998a) [26]
CYP716A52v2(OAS)	JX036032	F: AGGAGCAAATGGAGATAG R: AACCGTTGTAGGTGAAAT	Han et al. (2013) [28]
CYP716A47(PPDS)	JN604537	F: TCACCTTCGTTCTCAACTATC R: TCTTCCTCAAATCCTCCCAAT	Han et al. (2011) [27]
CYP716A53v2(PPTS)	JX036031	F: ATCGGACAACGAGGCAGCAC R: GCCAACAGGCCAACTCAA	Han et al. (2012) [29]

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
