# Peer review of "The Effects of Environmental Factors on Ginsenoside Biosynthetic Enzyme Gene Expression and Saponin Abundance"

_molecules, 2018, doi:10.3390/molecules24010014_

Round 1

Reviewer 1 Report

Comment to the Authors;

Dear Editor,

Here are my comments on this manuscript.

This manuscript titled “The effects of environmental factors on ginsenoside biosynthetic enzyme gene expression and saponin abundance” explains the relationship between environmental and genetic factors on the quality of the major compound ginsenoside contained in ginseng and some medicinal plant materials. Commendably, this research was carefully conducted at different growth stages. However, the authors should check and revise the following

1.    Grammatical errors and phrases. Please thoroughly review!

2.    Authors should check the journal guidelines on the arrangement of the manuscript based on; introduction, materials and methods and discussion.

3.     In line 38 “improve immunity, to inhibit apoptosis, and to help combat cardiovascular and cerebrovascular disease [5-11]”. Please check “to inhibit apoptosis”.

4.     In line 84-87. “The response patterns of ginsenoside content and gene 85 expressions to ecological factors were analyzed. To clarify the physiological and ecological mechanism 86 of ginsenoside biosynthesis, in order to provide theoretical support for guiding the cultivation of ginseng and the quality of 87 ginseng medicinal materials”. If authors intend to state this as an objective, it should be stated clearly

5.    Statistical significance should be included in figures 2, 3, 4,5,6,7 and 8. In the case of table 1 and 2 if possible LSD values should be included.

6.    Referencing should be rechecked. For example in line 347 “stored in the roots, as shown by Lee S W et al[53]” however, this is not the same as indicated in the reference list “52. Lee S W, Kang S W, Seong N S, et al. Seasonal Changes of Growth and Extract Content of Roots in Panax 572 Ginseng C.A.Meyer[J]. Korean Journal of Medicinal Science, 2004,6(12):483-489”.

Author Response

Dear Reviewers:

Thank you for your letter and for the reviewers’ comments concerning our manuscript entitled the effects of environmental factors on ginsenoside biosynthetic enzyme gene expression and saponin abundance. Those comments are all valuable and very helpful for revising and improving our paper, as well as the important guiding significance to our researches. We have studied comments carefully and have made correction which we hope meet with approval. The main corrections in the paper and the responds to the reviewer’s comments are as following:

Point 1:Grammatical errors and phrases. Please thoroughly review!

Response 1:  For the grammatical errors and phrases. I have undergone English language editing by MDPI. The text has been checked for correct use of grammar and common technical terms, and edited to a level suitable for reporting research in a scholarly journal. The relevant certificates are attached.

Point 2: Authors should check the journal guidelines on the arrangement of the manuscript based on; introduction, materials and methods and discussion.

Response 2:  I have checked the journal guidelines on the arrangement of the manuscript based on; introduction, materials and methods and discussion.

Point 3: In line 38 “improve immunity, to inhibit apoptosis, and to help combat cardiovascular and cerebrovascular disease [5-11]”. Please check “to inhibit apoptosis”.

Response 3: In line 38, I changed the “improve immunity, to inhibit apoptosis, and to help combat cardiovascular and cerebrovascular disease [5-11]” to “improve immunity, improve erectile function, and to help combat cardiovascular and cerebrovascular disease [5-11]”.

Point 4: In line 84-87. “The response patterns of ginsenoside content and gene 85 expressions to ecological factors were analyzed. To clarify the physiological and ecological mechanism 86 of ginsenoside biosynthesis, in order to provide theoretical support for guiding the cultivation of ginseng and the quality of 87 ginseng medicinal materials”. If authors intend to state this as an objective, it should be stated clearly.

Response 4:  In line 84-87. I have changed the sentence to "To explore the response patterns of ginsenoside biosynthesis and gene expression to ecological factors. It provides theoretical basis for elucidation of physiological and ecological mechanism of ginsenoside biosynthesis."

Point 5:   Statistical significance should be included in figures 2, 3, 4,5,6,7 and 8. In the case of table 1 and 2 if possible LSD values should be included.

Response 5:  Thank you for your suggestion, I have added statistical significance in Figures 2, 3, 4, 5, 6, 7 and 8. The LSD values of Table 2 have also been added. The modified image is visible in the revised manuscript. The data included in Table 1 is the ratio of the average value of PPD type ginsenoside to the average value of PPT type ginsenoside, the ratio of Rg1 to total saponin, the ratio of Re to total saponin, and the ratio of Rb1 to total saponin. By reading the relevant references[1], we find that most of papers and books do not add LSD values when writing the ratio of ginsenosides. Anyway, we thank for the reviewer's thoughtful advice.

 [1]        Kim Y, Jeon J, Jang M, et al. Ginsenoside profiles and related gene expression during foliation in Panax ginseng Meyer[J]. Journal of Ginseng Research, 2014,38(1):66-72.

Point 6Referencing should be rechecked. For example in line 347 “stored in the roots, as shown by Lee S W et al[53]” however, this is not the same as indicated in the reference list “52. Lee S W, Kang S W, Seong N S, et al. Seasonal Changes of Growth and Extract Content of Roots in Panax 572 Ginseng C.A.Meyer[J]. Korean Journal of Medicinal Science, 2004,6(12):483-489”.

Response 6:  The missing reference has been added into the revised manuscript and the references have been processed accordingly.

Reviewer 2 Report

In this paper, the authors study the biosynthesis of ginsenosides by Panax ginseng C.A. Meyer, which is regulated by environmental factors, via HPLC and real-time PCR. They correlated the expression of key genes with ecological factors. The analysis showed that in root tissue, PAR and soil water potential had a greater impact on ginsenoside accumulation, while in leaf tissue, temperature relative humidity had a greater impact on ginsenoside accumulation. Their results provide a basis for elucidating the relationship between ecological factors and genetic factors and the quality of medicinal materials.

The Authors should revise the English form throughout all the manuscript.

In the Abstract, the sentence in line 17-20 is repeated;

Page 2, lines 60 - 74: the molecular pathway here described is difficult to follow, they should integrate with the part in page 6, lines 159 – 169;

Page 2, lines 84-87 please, rephrase the sentence;

Page 4, lines 131-134 have been already mentioned above;

In the Discussion section, the authors should improve the explanation about  the significance of their results: in this section they repeated the description of the results, I suggest to make a change of that.

In addition they should better correlate the results obtained by other cited authors and not just to list them.

Author Response

Dear Reviewers:

Thank you for your letter and for the reviewers’ comments concerning our manuscript entitled the effects of environmental factors on ginsenoside biosynthetic enzyme gene expression and saponin abundance. Those comments are all valuable and very helpful for revising and improving our paper, as well as the important guiding significance to our researches. We have studied comments carefully and have made correction which we hope meet with approval. The main corrections in the paper and the responds to the reviewer’s comments are as following:

Point 1: The Authors should revise the English form throughout all the manuscript.

Response 1:   I follow the journal requirements, and I have revised the English form in the revised manuscript.

Point 2: In the Abstract, the sentence in line 17-20 is repeated.

Response 2: I have deleted the repeated sentences in the abstract of line 17-20.

Point 3: Page 2, lines 60 - 74: the molecular pathway here described is difficult to follow, they should integrate with the part in page 6, lines 159 – 169.

Response 3: I have combined page 2, lines 60-74 with page 6, lines 159-169 to describe the molecular pathways in the revised manuscript.

Point 4: Page 2, lines 84-87 please, rephrase the sentence.

Response 4In line 84-87. I have changed the sentence to "To explore the response patterns of ginsenoside biosynthesis and gene expression to ecological factors. It provides theoretical basis for elucidation of physiological and ecological mechanism of ginsenoside biosynthesis."

Point 5:   Page 4, lines 131-134 have been already mentioned above.

Response 5I have deleted page 4, lines 131-134 that have been already mentioned above, in the revised manuscript.

Point 6:  In the Discussion section, the authors should improve the explanation about the significance of their results: in this section they repeated the description of the results, I suggest to make a change of that. In addition they should better correlate the results obtained by other cited authors and not just to list them.

Response 6:  In the discussion section of the revised manuscript, I have further explained and analyzed the results. And correlate it better with the results obtained by the cited authors. I have changed this part:

In this study, PAR was significantly positively correlated with Rg1, Rf and PPT-type ginsenosides. The ginsenoside content of ginseng roots increased as the light transmission rate (LTR) increased. A study of 2-year-old ginseng reported that the total ginsenoside content of ginseng grown at 17% LTR was 49.7% and 68.3% higher than ginseng grown at 6% LTR in August and at final harvest, respectively[38]. The results are consistent with the results of this study. This indicated that appropriate PAR was beneficial to the accumulation of ginsenosides. In the leaves, relative humidity was significantly positively correlated with all ginsenosides, except for the Ro and PPT-type ginsenosides. Relative humidity can appropriately alleviate the drought limit of the soil, which is conducive to plant leaf growth and life activities. Ginseng leaves are sensitive to light, and excessive photosynthetically active radiation may damage the leaves, thus causing the reduction of ginsenosides. Soil water potential were negatively correlated with most of the individual ginsenosides, and suitable drought can improve the biosynthesis of ginsenosides. Relative humidity is an important factor affecting the quality of the medicinal materials. Appropriate drought stress can improve the active ingredients in the medicinal materials[39].

With the development of molecular biology, people also focus on the influence of intrinsic genetic factors on the formation of medicinal materials[40]. HMGR was differentially expressed among tissues, with high level of expression in the leaf and low level of expression in the stem, suggesting that leaves are crucial to terpenoid biosynthesis[41]. Kim et al. reported that overexpression of FPS caused an approximate 2.4-fold increase of ginsenoside content in transgenic ginseng hairy roots[42]. Seo et al introduced the SS gene of P. ginseng into Eleutherococcus senticosus by Agrobacterium-mediated transformation, that SS activity significantly improved the production of phytosterols and triterpenoids[43]. He et al cloned SQE from the root of P. notoginseng by PCR. Real time quantitative PCR analysis showed that its cDNA had different expression pattern and is highly expressed in root, especially in three-year-old root[44]. Chen et al analyzed the transcriptomes of P. ginseng and identified 133 CYP450 genes by 454 sequencing technology. Their study laid an important foundation for the further screening of CYP450 involved in ginsenoside biosynthesis[45].

In this study, expression analysis by real-time quantitative PCR indicated that genes were differentially expressed among tissues. SS, DS, SE, PPTS and PPDS with high level of expression in the leaf, this is closely related to the distribution of dammarane ginsenoside in leaves, and FPS, HMGR, β-AS and OAS with high level of expression in the root, this is consistent with the distribution of Ro mainly in the root. Through the analysis of the gene correlation, it was found that there was a significant correlation between the expression of multiple enzyme genes in the pathway, indicating that multiple ginsenoside synthesis genes in the pathway interacted or had the same regulatory element, which was a key gene in ginsenoside biosynthesis. Through correlation analysis of gene expression and saponins content, it was found that multiple enzymes in the pathway were significantly correlated with single saponins at different growth stages, and thus played different regulatory roles.

[38]  Jang I, Dae-Young Lee, Jin Yu, et al. Photosynthesis rates, growth, and ginsenoside contents of 2-yr-old Panax ginseng grown at different light transmission rates in a greenhouse[J]. Journal of Ginseng Research, 2015,39(1):345-353.

[39]  Z S, S C, M L. Physiological and Biochemical Response of Panax ginseng C.A. Meyer to Drought Stress[J]. Journal of Northeast Agricultural Sciences, 2016(01).

[40]  Yang J, Hu Z, Zhang T, et al. Progress on the Studies of the Key Enzymes of Ginsenoside Biosynthesis[J]. Molecular, 2018,589(23):1-12.

[41]  Qiong W, Chao S, ShiLin C. Identification and expression analysis of a 3-hydroxy-3-methylglutaryl coenzyme A reductase gene from American ginseng.[J]. Plant Omics, 2012,4(5):414-420.

[42]  Kim Y K, Kim Y B, Uddin M R, et al. Enhanced triterpene accumulation in Panax ginseng hairy roots overexpressing mevalonate-5-pyrophosphate decarboxylase and farnesyl pyrophosphate synthase[J]. ACS Synth Biol, 2014,3(10):773-779.

[43]  Seo J W, Jeong J H, Shin C G, et al. Overexpression of squalene synthase in Eleutherococcus senticosus increases phytosterol and triterpene accumulation[J]. Phytochemistry, 2005,66(8):869-877.

[44]  He F, Zhu Y, He M, et al. Molecular cloning and characterization of the gene encoding squalene epoxidase in Panax notoginseng[J]. DNA Seq, 2008,19(3):270-273.

[45]  Chen S, Luo H, Li Y, et al. 454 EST analysis detects genes putatively involved in ginsenoside biosynthesis in Panax ginseng[J]. Plant Cell Rep, 2011,30(9):1593-1601.

Round 2

Reviewer 2 Report

The new version of the manuscript meets the referee's request and may be published as it is.